# Investigating the role of perceived autonomy support in moderating the association between diabetes stigma and psychological and diabetes self-management outcomes among adults with type 2 diabetes in Ghana

Samuel Akyirem[1]*, Katie Wang[2], Gail Melkus[3], Soohyun Nam[1], Frank Micah[4], Emmanuel Ekpor[5,6], LaRon E. Nelson[1]

1 Yale University, School of Nursing, Orange, Connecticut, United States of America, 2 Yale School of Public Health, New Haven, Connecticut, United States of America, 3 New York University, New York, New York, United States of America, 4 Komfo Anokye Teaching Hospital, Kumasi, Ghana, 5 School of Psychology | Institute for Health Transformation, Deakin University, Geelong, Victoria, Australia, 6 The Australian Centre for Behavioral Research in Diabetes, Diabetes Victoria, Carlton, Victoria, Australia

* samuel.akyirem@yale.edu

## Abstract

Studies on diabetes-related stigma rarely focus on identifying or examining protective factors (e.g., social support and healthcare environment) that can mitigate the adverse effect of this social phenomenon. In this cross-sectional study, we examined perceived autonomy support, a concept from the self-determination theory, as a moderator of the association between diabetes-related stigma and its adverse behavioral and psychological (depression, diabetes distress, and anxiety) correlates. We recruited 190 adults with type 2 diabetes (T2D) from a tertiary hospital in Ghana. We assessed diabetes-related stigma, depression, anxiety, diabetes distress, diabetes self-management, and perceived autonomy support using psychometric instruments. Hierarchical multivariable linear regressions were used to evaluate moderation effects of perceived autonomy support. Participants had an average age of 59.44 (standard deviation [SD] = 10.7) years, were mostly female (70.5%, n = 134), and had had T2D diagnosis for a median of 14.5 years. Autonomy support was directly associated with lower anxiety and depression and better diabetes self-management behaviors. Greater perceived autonomy support reduced the negative association between diabetes-related stigma and diabetes self-management (β = 0.20, 95% confidence interval [CI]: 0.01 to 0.39; p = 0.041). Perceived autonomy support buffered the negative effects of diabetes stigma on self-management. These findings highlight autonomy-supportive care as a promising strategy to address the adverse effects of diabetes-related stigma in Ghana.

**Data availability statement:** All data can be found in the manuscript and supporting information files.

**Funding:** The study was funded by Sigma Theta Tau International (Small Grants), Whitney and Betty MacMillan Center for International and Area Studies at Yale (International Dissertation Fellowship and Lindsay African Fellowship), and Yale Women Faculty Forum (Seed Grant). The funders had no role in study design, data collection and analysis, decision to publish, or preparation of the manuscript.

**Competing interests:** The authors have declared that no competing interests exist.

## Introduction

The International Diabetes Federation estimates that about 24 million adults in Africa have diabetes as of 2021, a number that is expected to increase by 129% to 55 million in 2045 [1]. In Ghana, the Global Burden of Disease study estimates that 5.3% of Ghanaians have diabetes, with type 2 diabetes (T2D) being the most common form of the disease [2]. Adults living with T2D are expected to play an active role in their disease management, including implementing significant lifestyle changes such as engagement in regular physical activity, healthy eating, and taking medications [3]. These lifestyle changes following diabetes diagnosis can be physically and psychosocially daunting, and adults with T2D may benefit from support from family, friends, and healthcare professionals [4].

In addition to lifestyle changes, adults with T2D experience diabetes-related stigma in the form of being blamed, judged, and discriminated against because of their diabetes [5]. In Ghana, diabetes-related stigma may emerge from illness interpretations in which visible symptoms of diabetes, such as sudden weight loss in the early stages of diagnosis and delayed wound healing at later stages, are socially interpreted through moralized or spiritual lens, leading to misidentification with other highly stigmatized conditions (e.g., HIV), accusations of witchcraft, and, consequently, social ostracism [6,7]. Other studies have also reported diabetes-related stigma in healthcare settings. A study among adults with T2D in Australia found that over 15% of participants reported experiencing some form of diabetes-related stigma in the healthcare settings (e.g., being treated with less respect because of one's diabetes) [8]. This is striking given the critical role healthcare professionals play in educating, supporting, and empowering adults with T2D [9].

Diabetes-related stigma can impact psychological, behavioral, and medical health outcomes [5]. Evidence suggests that diabetes-related stigma is significantly associated with avoidance of self-management behaviors, poor glycemic management, and adverse psychological outcomes [10,11]. For instance, a recent systematic review and meta-analysis reported a small-to-medium positive correlation between diabetes-related stigma and depressive symptoms, anxiety symptoms, and diabetes distress (often defined as the negative emotions that result from the daily challenge of dealing with the demands of diabetes) [12,13].

Whereas several studies have highlighted the negative effect of diabetes-related stigma on health outcomes, fewer studies have examined the role of protective factors in mitigating the adverse effects of stigma. Studies have shown that factors such as resilience, social support, and self-esteem can be protective of the health effect of diabetes stigma [10,14,15]. Perception of autonomy support is one protective factor that has not been investigated in the diabetes-related stigma literature, to the best of our knowledge, but has shown great promise in other health related stigmas including HIV stigma [16]. The concept of autonomy support is drawn from the self-determination theory (SDT). According to the SDT, individuals' motivation is driven by three basic psychological needs: autonomy, competence, and relatedness [17,18]. The psychological need for autonomy indicates the need to feel that one is in control

of their choices and actions. SDT posits that when one's external environment is supportive of their autonomy, competence, and relatedness, one's actions towards healthy behavior such as engaging in physical activity become automatic as the individual becomes intrinsically motivated [18].

Autonomy support is defined as the degree to which healthcare providers consider and include patients' perspectives in their care and provide the necessary resources needed by patients to make informed choices about their own health while making patients feel accepted regardless of their health choices [19]. Autonomy supportive healthcare environment emphasizes patient-centered care and has been shown to improve psychological outcomes and the adoption of healthy lifestyle changes including physical activity [20,21] that improve glycemia of adults with T2D [19]. Additionally, health interventions that emphasize autonomy support have shown promise in improving physical and psychological outcomes among adults living with chronic diseases [22]. It is, however, not clear if perceived autonomy support plays any protective role in mitigating the adverse effects of diabetes stigma. Additionally, it is not clear whether autonomy support in the healthcare environment is beneficial in a highly hierarchical collectivist society like Ghana, in which individuals are more likely to defer decision-making to the elderly and people in authority including health professionals [23]. Unlike Western cultures, which often emphasize individualism and self-determination, collectivist societies may hold different notions of personal autonomy, placing greater emphasis on individuals' obligations to family and community [24,25]. As a result, autonomy is a more nuanced construct in this cultural context and may not manifest in the same way as it does in more individualistic societies. Whereas autonomy support is universal, it may not present in the same manner and may not be as beneficial across cultures [26,27], hence the need to study this phenomenon in diverse cultural settings.

The purpose of this study was to assess perceived autonomy support as a moderator of the association between diabetes-related stigma and psychological and self-management outcomes among adults with T2D in Ghana, a collectivist society. The study tested the hypotheses that 1) perceived autonomy support is significantly associated with better psychological outcomes (lower anxiety symptoms, diabetes distress, and depressive symptoms) and self-management behaviors, 2) perceived autonomy support moderates the association between diabetes stigma and psychological outcomes, such that higher autonomy support will buffer the association between diabetes stigma and psychological outcomes, and 3) perceived autonomy support moderates the association between diabetes stigma and self-management, such that higher autonomy support will weaken the negative association between diabetes stigma and diabetes self-management behaviors.

## Materials and methods

This study was an analysis of data from a cross-sectional study of adults living with T2D. Study participants were drawn from the outpatient diabetes clinic at the Komfo Anokye Teaching Hospital (KATH) in Kumasi, Ghana. KATH is a tertiary hospital in the southern part Ghana. The hospital serves as the primary referral point for all healthcare facilities within 10 out of Ghana's 16 administrative regions and other neighboring countries [28]. The outpatient diabetes clinic has a weekly attendance of between 300 and 450 patients [28].

### Eligibility criteria

The inclusion criteria were: 1) adults ≥18 years; 2) diagnosis of T2D as indicated in participants' health records; 3) living with T2D for at least 1 year; 4) ability to read, speak, and/or understand English or Twi (the most popular local dialect in Ghana) [29,30]. Exclusion criteria included: 1) severe neurological diseases (such as dementia) characterized by severe cognitive decline that may impair participant's ability to provide consent and complete study questionnaires, 2) conditions that may impact hemoglobin A1C (HbA1c) assessment including pregnancy, breastfeeding, receiving treatment for severe anemia, glucose-6-phosphate dehydrogenase deficiency, end-stage kidney disease, taking steroids for a long-term condition (including cancer), bariatric surgery in the past 3 months, and experiencing major life events (such as death of a close friend/family/spouse, major injuries/illness, and retirement/job loss) in the past 3 months.

## Study procedures

Participants recruitment and data collection occurred between July 3rd and September 6th, 2024. Participants were selected using both purposive and convenient sampling approaches. These sampling strategies allowed for recruitment of eligible participants who were readily accessible during the data collection period. Four trained research assistants (RAs) supported participant recruitment and data collection. The RAs were either registered nurses or nursing students. They were trained in the study procedures, consent processes, and the use of Qualtrics to administer study questionnaire and record participants' responses. The RAs approached adults with T2D who were waiting to be seen by their provider at the outpatient diabetes clinic at KATH. The RAs made announcements to all patients in the waiting area, providing details about eligibility for the study, study procedures, and study duration. Participants who were interested in the study approached the research team for detailed explanation of study procedures and formal assessment of eligibility (convenient sampling). Those who agreed to participate were then asked to complete the informed consent form. It should be noted that the first author [SA] regularly reviewed the demographic characteristics (gender, education, and age) of recruited participants and adjusted the recruitment strategy to target under-represented groups (purposive sampling). This was done to increase the diversity of the study sample and to ensure that the distribution of demographic characteristics in our sample was similar to the T2D population who access care at KATH [31].

Survey questionnaires were researcher-administered (either by the first author [SA] or the trained RAs) with the option for participants to self-administer in a private space. No participant chose to self-administer. For participants who were not comfortable communicating in English, we verbally read the questionnaire items and responses in the Twi language and recorded participants' responses accordingly. All questionnaires were administered via the Qualtrics offline mobile app. Participants were compensated with 350 Ghanaian Cedis (~30 USD) after completing all study procedures. While participant compensation is sometimes discussed as a potential source of selection bias [32], the amount provided in our study was modest, ethically approved, and intended to offset time and transportation costs rather than serve as an inducement. As such, it is less likely that compensation meaningfully influenced participation decisions.

## Measures

**Perception of autonomy support.** The brief version of the Health Care Climate Questionnaire (HCCQ) was used to assess perception of autonomy support [33,34]. The brief HCCQ measures participants' perception of the level of autonomy support they received from their healthcare providers using 6 items. Each item on the instrument is rated on a 7-point scale ranging from 1 (strongly disagree) to 7 (strongly agree). Representative items on the scale include: "My physician conveys confidence in my ability to make changes" and "My physician listens to how I would like to do things". The total score is calculated by averaging across all items. A higher HCCQ score reflects greater perceived autonomy support in the healthcare environment. The scale demonstrated good psychometric properties with Cronbach's alpha of 0.82 in the current study.

**Diabetes stigma.** Diabetes stigma was measured using the Type 2 Diabetes Stigma Assessment Scale (DSAS-2) [8]. The 19-item DSAS-2 scale consists of 3 subscales: blame and judgement, being treated differently, and self-stigma. Previous studies have confirmed that the scale has adequate convergent, concurrent and discriminant validity [8]. Moreover, the scale has been translated and validated in multiple countries [12]. Each DSAS-2 item is assessed on a 5-point Likert scale: 1 = "strongly disagree", 2 = "disagree", 3 = "unsure", 4 = "agree", and 5 = "strongly agree". Some items on the DSAS-2 include "I feel embarrassed because of my type 2 diabetes" and "Health professionals think that people with type 2 diabetes don't know how to take care of themselves". The total score of the overall scale and subscales are calculated by summing the scores of individual items (range: 19–95). Higher DSAS-2 scores indicate higher level of T2D stigma. For this study, we reported internal consistency of 0.83.

**Depressive symptoms.** Depressive symptoms was assessed by the 8-item Patient-Reported Outcomes Measurement Information System (PROMIS) Depression Short Form 8a [35]. The PROMIS depression scale measures depression in the past 7 days using a 5-point Likert scale ranging from 1 = "never" to 5 = "always". Some items include: "In the past 7 days, I felt like a failure" and "In the past 7 days, I felt sad". The total raw depressive symptoms score is obtained by summing the scores for each item. Higher scores reflect worse depressive symptoms. The raw scores were then converted to T-scores where 50 and 10 represent the mean and standard deviation (SD) of the reference population respectively. T-scores for this scale range from 8 to 81.1. The T-scores, in lieu of the raw scores, were used for all subsequent analyses.

**Anxiety symptoms.** Anxiety symptoms was assessed by the 8-item PROMIS Anxiety Short Form 8a [36]. The PROMIS anxiety instrument measures the level of anxiety in the past 7 days. Some items on the scale include "In the past 7 days, I felt uneasy" and "In the past 7 days, I felt tense". Each item is rated on a 5-point Likert scale ranging from Never (1), Rarely (2), Sometimes (3), Often (4), and Always (5). The total raw anxiety score is obtained by summing the scores for each item. Higher scores reflect higher intensity of anxiety symptoms. The raw anxiety score is often re-scaled to a T-score where 50 and 10 represent the mean and standard deviation (SD) of the referent population respectively. T-scores for the anxiety scale range from 8 to 83.1. The T-scores, in lieu of the raw scores, were used for all subsequent analyses.

**Diabetes distress.** Diabetes-related distress was measured by the Problem Areas in Diabetes Scale (PAID) [37]. The PAID has 20 items. Each item is scored on a five-point Likert scale ranging from 0 to 4 with 0 representing "not a problem" and 4 "a serious problem" [37]. Representative items on the scale include "worrying about low blood glucose reactions?" and "Not accepting your diabetes?". The scores for each item are summed and multiplied by 1.25, to obtain a total score that ranges from 0 to 100. Participants scoring 40 or higher may be classified as having "severe diabetes distress" [38]. PAID has demonstrated good psychometric properties including a Cronbach's alpha of 0.83 among adults with T2D in Ghana [39]. In the current study, Cronbach's alpha was 0.90.

**Diabetes self-management.** Diabetes self-management was assessed by the Diabetes Self-management Questionnaire (DSMQ) [40]. The DSMQ has 16 items grouped under 4 subscales: "Glucose Management" (5 items), "Dietary Control" (4 items), "Physical Activity" (3 items), and "Health-Care Use" (3 items). Items are rated on a 4-point Likert scale 0 = "does not apply to me" to 3 = "applies to me very much". Some items on the scale are: "Sometimes I have real 'food binges' (not triggered by hypoglycemia)" and "I do regular physical activity to achieve optimal blood sugar levels". The overall DSMQ score is obtained by summing all items on the instrument and transforming the score to a 0–10 scale using the formula:

$$(actual\ sum\ of\ items/maximum\ possible\ sum\ of\ items)\ x\ 10.$$

DSMQ has good convergent validity as demonstrated by its correlation with the Summary of Diabetes Self-Care Activities scale (SDSCA) [40]. Higher scores indicate higher engagement in self-management behaviors.

**Sociodemographic and clinical variables.** The sociodemographic variables included age, gender, income, employment, education, marital status, and place of residence. Clinical variables included duration of T2D diagnosis, insulin use, and family history of diabetes. These variables were selected because of their known associations with diabetes stigma and outcomes [41–46].

## Ethical approval

We obtained Institutional Review Board (IRB) approval for this study from Yale University (IRB# 2000036937, approved on February 7, 2024) and KATH (IRB# KATH/IRB/AP/031/24; approved on April 9, 2024). All participants provided written informed consent prior to data collection. Participants' data were uploaded to Yale University Qualtrics servers daily during

the data collection period. De-identified data were eventually downloaded and stored on password-protected Yale University OneDrive servers ensuring that participants confidentiality and data security were maintained throughout the process.

## Inclusivity in global research

Additional information regarding the ethical, cultural, and scientific considerations specific to inclusivity in global research is included in the Supporting Information (S1 Checklist).

## Statistical analysis

Data analysis was conducted with R statistical software [47]. Descriptive statistics including means, standard deviations, counts, and proportions were used to summarize the data. We examined the bivariate associations among all continuous variables using Pearson correlation. Four hierarchical multivariable linear regression models with robust standard errors were used to test our hypotheses (models 1–4). Hierarchical models were used to examine the incremental contribution of key independent variables to the outcome. Prior to the regression, diabetes-related stigma (DSAS-2) and perceived autonomy support (HCCQ) scores were standardized using z-scores to reduce multicollinearity of interaction term. Each regression model included two variable blocks. Block 1 included ten sociodemographic and clinical covariates (e.g., insulin use and age) as well as DSAS-2 and HCCQ z-scores to examine main effects of perceived autonomy support and diabetes-related stigma. Block 2 included an interaction term between DSAS-2 and HCCQ variables in addition to variables in block 1. Outcome variables for models 1,2,3 and 4 were diabetes self-management, depressive symptoms, anxiety symptoms, and diabetes distress respectively. Significance ($p < 0.05$) of model blocks and interaction terms was assessed with F statistics and regression coefficients respectively. We also reported $\Delta R^2$ for each block of variables added to the model. For outcomes where there were significant interaction effects, we used the Johnson-Neyman (JN) technique to visualize the moderating effects of HCCQ and identify the ranges of HCCQ at which the association between diabetes stigma and outcomes were significant [48]. The JN technique, also known as "flood light analysis", was chosen because it examines the conditional effect across the full range of the moderator, eliminating the need for arbitrary cut-points such as one standard deviation above and below the mean [48]. For all inferential analyses, missing data was handled through listwise deletion. No data imputation was performed. All p-values of 0.05 or less were considered statistically significant.

## Results

### Characteristics of study participants

Table 1 shows the sociodemographic and clinical characteristics of the 190 participants included in the study. Participants had an average age of 59.44 (SD = 10.7) years, were mostly under 55 years (38.4%, n = 73), women (70.5%, n = 134), had up to Junior High School education (62.6%, n = 119), earned 1500 or less Ghanaian Cedis. Majority of participants were married or living with a partner (59.5%, n = 113) and employed (52.6%, n = 100). The median duration of T2D diagnosis was 14.5 years (range: 1–55 years). Four-fifth of study participants (80%, n = 152) had a family history of T2D. A little over one-third (36.3%, n = 69) were on daily insulin injections at the time of data collection. Majority of the participants were living with comorbidities (78.9%, n = 150) with most having hypertension (n = 133, 70%), followed by eye problems (n = 72, 37.9%).

### Bivariate associations among study variables

Table 2 shows descriptive statistics and bivariate associations among study variables. Participants had a mean DSAS-2 score of 41.80 (10.7) with 11.1% demonstrating "problematic stigma" (i.e., scoring one standard deviation above the mean, a definition recommended by the original developers of the DSAS-2 scale [8]). About a quarter of the participants had moderate-to-severe diabetes distress (27.12%). About 28.5% and 35.6% of participants had depression and anxiety respectively (i.e., having T-scores > 55).

**Table 1. Sociodemographic and clinical characteristics (N = 190).**

| Characteristics | n (%) |
|---|---|
| Gender | |
| Male | 56 (29.5) |
| Female | 134 (70.5) |
| Age (mean, SD) | *59.44 (10.7)* |
| <55 | 73 (38.4) |
| 55-65 | 58 (30.5) |
| >65 | 59 (31.1) |
| Education | |
| No formal education | 15 (7.9) |
| Up to Junior High School | 119 (62.6) |
| Up to Senior High School | 30 (15.8) |
| At least college | 26 (13.7) |
| Monthly income | |
| 1500 cedis or less | 112 (58.9) |
| 1501–3000 cedis | 49 (25.8) |
| >3000 cedis | 27 (14.2) |
| Missing | 2 (1.1) |
| Marital | |
| Married or living with partner | 113 (59.5) |
| Divorced | 27 (14.2) |
| Never married | 6 (3.2) |
| Separated | 3 (1.6) |
| Widowed | 41 (21.5) |
| Employment | |
| Employed (private, public, self-employed) | 100 (52.6) |
| Unemployed | 42 (22.1) |
| Not in the workforce (student, retired, homemaker) | 48 (25.3) |
| Time to reach nearest health facility | |
| <15 minutes | 90 (47.4) |
| 15 – 30 minutes | 63 (33.1) |
| 31 – 45 minutes | 15 (7.9) |
| >= 46minutes | 22 (11.6) |
| Insulin use | |
| Yes | 69 (36.3) |
| No | 121 (63.7) |
| Duration of diabetes diagnosis (median, range) | 14.5 (range 1–55) years |
| Comorbidities (hypertension, heart disease, eye problems, HIV, and kidney disease) | |
| None | 40 (21.1) |
| At least 1 comorbidity | 150 (78.9) |
| Family history of diabetes | |
| Yes | 152 (80.0) |
| No | 38 (20.0) |

**Table 2. Descriptive statistics and bivariate associations among autonomy support, psychological, and self-management outcomes.**

|  | Mean | SD | 1 | 2 | 3 | 4 | 5 | 6 |
|---|---|---|---|---|---|---|---|---|
| 1. Diabetes stigma | 41.80 | 10.17 | 1 |  |  |  |  |  |
| 2. Self-management | 7.01 | 1.32 | -0.27** | 1 |  |  |  |  |
| 3. Anxiety symptoms | 51.50 | 9.72 | 0.58*** | -0.33*** | 1 |  |  |  |
| 4. Depressive symptoms | 50.62 | 9.11 | 0.57*** | -0.35*** | 0.81*** | 1 |  |  |
| 5. Diabetes distress | 28.77 | 19.23 | 0.51*** | -0.18* | 0.61*** | 0.56*** | 1 |  |
| 6. Autonomy support | 5.54 | 1.09 | -0.12 | 0.24** | -0.23** | -0.27*** | -0.09 | 1 |

There was a significant correlation among all study variables except for perceived autonomy support and diabetes distress and diabetes-related stigma. As shown in Table 1 and Fig 1, the strongest correlations were observed between depressive and anxiety symptoms (r = 0.81), and diabetes distress and anxiety symptoms (r = 0.61). Perceived autonomy support was weakly but statistically significantly associated with lower anxiety (r = −0.23) and fewer depressive symptoms (r = −0.27). Greater perceived autonomy support was also weakly associated with better diabetes self-management (r = 0.24).

**Results of the multivariable linear regression.** In Model 1, where diabetes self-management was an outcome, we found that perceived autonomy support was associated with better diabetes self-management (β = 0.28, 95% CI: 0.06 to 0.51; p = 0.013) whereas diabetes-related stigma was associated with worse diabetes self-management (β = -0.27, 95%CI: -0.46 to -0.08, p = 0.006). Block 1 variables accounted for 11.6% of the variance in diabetes self-management. When Block 2 variables were added, we observed a significant increase in the variance of diabetes self-management explained ($\Delta R^2$ = 0.02, p = 0.0285). In line with our hypothesis, perceived autonomy support moderated the association between diabetes-related stigma and diabetes self-management (β = 0.20, 95% CI: 0.01 to 0.39; p = 0.041). Specifically, greater perceived autonomy support reduced the negative association between diabetes-related stigma and diabetes self-management (Fig 2). The Johnson-Neyman plot (Figs 2 and 3) showed that perceived autonomy support had significant moderating effects on the stigma self-management association at HCCQ z-scores outside the range of 0.58 – 13.44 (Note: the range of observed HCCQ z-scores in our dataset was from -3.55 to 1.34).

In Model 2, depressive symptoms were the outcome. We found that greater perceived autonomy support was associated with lower depressive symptoms (β = -2.57, 95% CI: -3.72 to -1.42; p < 0.0001) whereas diabetes-related stigma was associated with higher depressive symptoms (β = 4.71, 95% CI: 3.58 to 5.84; p < 0.0001). Block 1 variables accounted for 44.1% of the variance in depressive symptoms. The addition of block 2 variables did not result in any significant change in the variance of depressive symptoms explained ($\Delta R^2$ = - 0.0001, p = 0.3922). Contrary to our hypothesis, perceived autonomy support was not a significant moderator of the association between diabetes-related stigma and depressive symptoms (β = 0.43, 95% CI: -0.73 to 1.59; p = 0.468).

Model 3 had anxiety symptoms as outcome. We found that greater perceived autonomy support was associated with lower anxiety symptoms (β = -2.19, 95% CI: -3.47 to -0.91; p = 0.001). Diabetes-related stigma was associated with greater anxiety symptoms (β = 5.16, 95% CI: 3.94 to 6.38, p < 0.0001). Block 1 variables accounted for 39.0% of the variance in anxiety symptoms. We did not find any significant change in the variance of anxiety symptoms explained when the interaction term was added ($\Delta R^2$ = - 0.003, p = 0.697). Contrary to our hypothesis, perceived autonomy support was not a significant moderator of the association between diabetes-related stigma and anxiety symptoms (β = 0.22, 95% CI: -0.95 to 1.39; p = 0.711).

Model 4 assesses diabetes distress as an outcome. We found no significant association between perceived autonomy support and diabetes distress (β = -1.23, 95% CI: -3.81 to 1.35; p = 0.352). Diabetes-related stigma, on the other hand was associated with higher diabetes distress (β = 9.05, 95% CI: 6.29 to 11.82, p < 0.0001). Block

**PLOS Global Public Health**

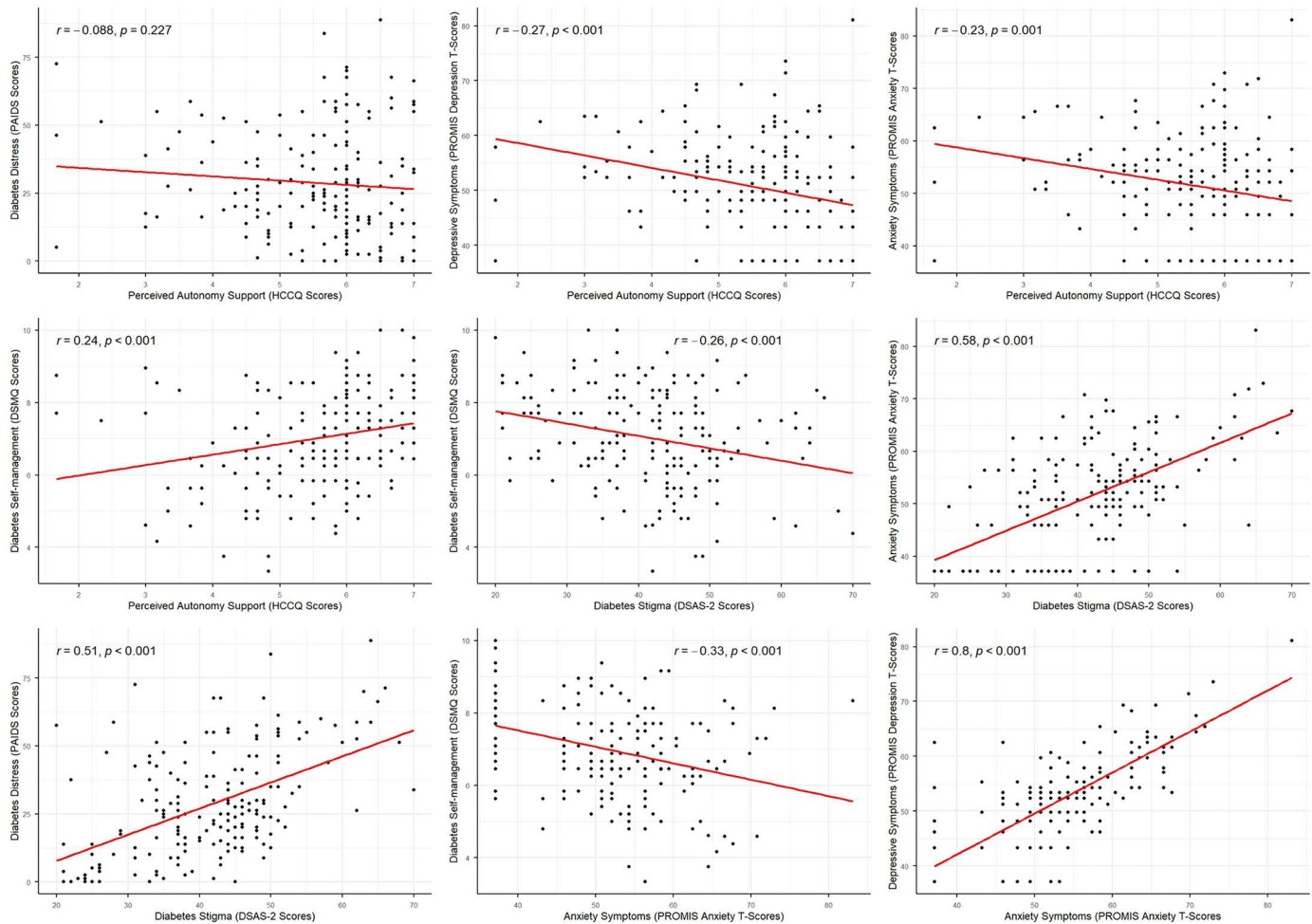

**Fig 1. Scatter plot showing the association among study variables.** We observe that perceived autonomy support is associated with lower depressive and anxiety symptoms and better self-management behaviors. The red trend line indicates direction of the correlation.

1 variables accounted for 34.1% of the variance in diabetes distress. The addition of block 2 variables containing interaction term did not result in any significant change in the variance of diabetes distress explained ($\Delta R^2 = -0.0004$, $p = 0.344$). In contrast to our *a priori* hypothesis, perceived autonomy support was not a significant moderator of the association between diabetes-related stigma and diabetes distress ($\beta = 1.09$, 95% CI: -1.08 to 3.26; $p = 0.327$) (Table 3).

## Discussion and conclusion

To the best of our knowledge, this study is the first to examine the potential protective effects of perceived autonomy support in the context of diabetes-related stigma. This is also one of the earliest studies on diabetes-related stigma in Africa. We found that greater perceived autonomy support reduced the negative association between diabetes-related stigma and diabetes self-management. Additionally, although, greater perceived autonomy support did not significantly moderate the stigma-psychological outcomes association, it demonstrated significant direct associations with lower anxiety and depressive symptoms in adults with T2D in Ghana.

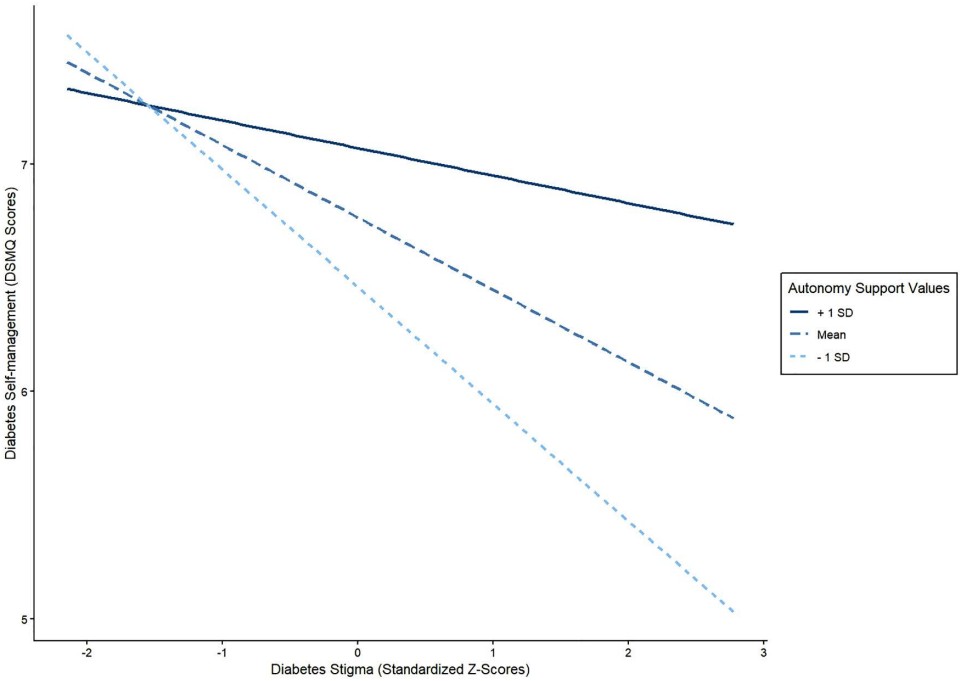

**Fig 2. Spotlight analysis of the effect of T2D stigma on diabetes self-management behaviors at three levels of perceived autonomy support (mean, mean + SD, and mean -SD).** At lower levels of perceived autonomy support (mean - SD), the negative effect of T2D stigma on self-management behaviors is steep compared to the mean autonomy support.

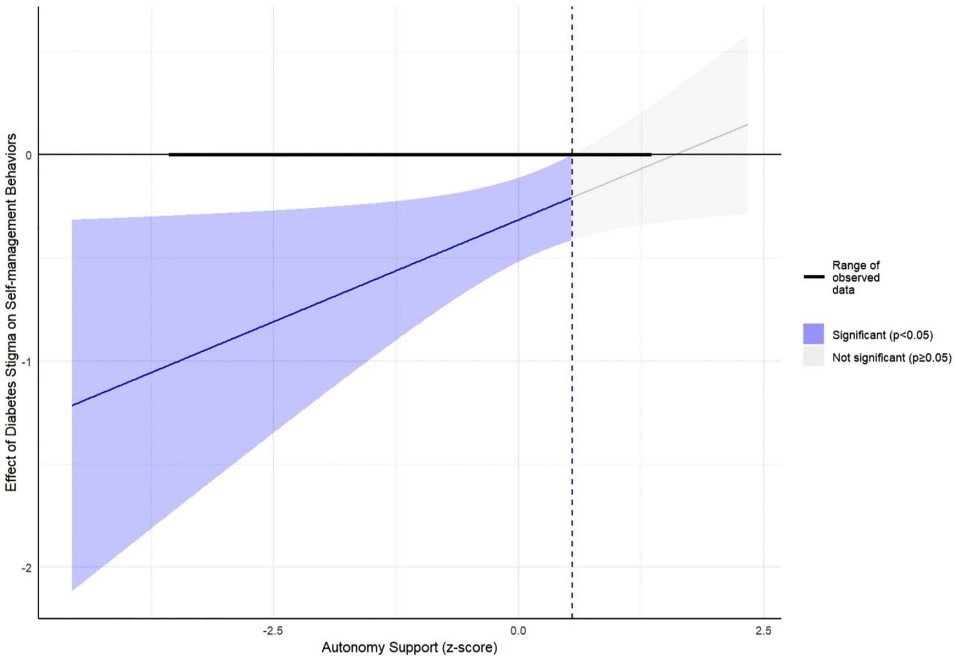

**Fig 3. Johnson-Neyman plot showing ranges of autonomy support for which the association between diabetes stigma and self-management behaviors is significant.** We observe that as autonomy support increases, the effect of diabetes stigma on self-management behaviors approaches zero.

Table 3. Hierarchical multivariable linear regression (models 1-4).

| Variables | Model 1: Self-management behaviors | | Model 2: Depressive symptoms | | Model 3: Anxiety symptoms | | Model 4: Diabetes distress | |
|---|---|---|---|---|---|---|---|---|
| | β [95% CI] | p-value | β [95% CI] | p-value | β [95% CI] | p-value | β [95% CI] | p-value |
| **Block 1** | | | | | | | | |
| DSAS-2 | -0.27 [-0.46, -0.08] | 0.006 | 4.71 [3.58, 5.84] | <0.0001 | 5.16 [3.94, 6.38] | <0.0001 | 9.05 [6.29, 11.82] | <0.0001 |
| HCCQ | 0.28 [0.06, 0.51] | 0.013 | -2.57 [-3.72, -1.42] | <0.0001 | -2.19 [-3.47, -0.91] | 0.001 | -1.23 [-3.81, 1.35] | 0.352 |
| Model fit | $R^2$ = 11.55%, $F_{19,167}$ = 2.279, p = 0.003 | | $R^2$ = 44.13%, $F_{190,166}$ = 8.692, p < 0.0001 | | $R^2$ = 38.98%, $F_{19,167}$ = 7.185, p < 0.0001 | | $R^2$ = 34.07%, $F_{19,165}$ = 6.004, p < 0.0001 | |
| **Block 2** | | | | | | | | |
| DSAS-2 | -0.32 [-0.51, -0.12] | 0.002 | 4.61 [3.47, 5.75] | <0.0001 | 5.10 [3.88, 6.33] | <0.0001 | 8.80 [6.09, 11.51] | <0.0001 |
| HCCQ | 0.31 [0.10, 0.52] | 0.005 | -2.53 [-3.72, -1.34] | <0.0001 | -2.17 [-3.45, -0.89] | 0.001 | -1.13 [-3.60, 1.34] | 0.371 |
| DSAS-2 X HCCQ | 0.20 [0.01, 0.39] | 0.041 | 0.43 [-0.73, 1.59] | 0.468 | 0.22 [-0.95, 1.39] | 0.711 | 1.09 [-1.08, 3.26] | 0.327 |
| Model fit | $R^2$ = 13.56%, $F_{20,166}$ = 2.459, p = 0.001 | | $R^2$ = 44.04%, $F_{20,165}$ = 8.281, p < 0.0001 | | $R^2$ = 38.66%, $F_{20,164}$ = 6.799, p < 0.0001 | | $R^2$ = 34.03%, $F_{20,164}$ = 5.746, p < 0.0001 | |

*All models are adjusted for insulin use, marital status, education, time since diabetes diagnosis, age, employment, gender, waist-to-height ratio (z-score), family history of diabetes, and income,*

*Abbreviations: DSAS-2 = Type 2 diabetes stigma assessment scale; HCCQ = healthcare climate questionnaire; CI = Confidence Interval.*

Perceived autonomy support, which emphasizes the quality of patient-provider interactions, plays an important role in how patients are motivated to engage in healthy behaviors for better health outcomes [49]. When healthcare professionals support patients' psychological need for autonomy through giving ample consideration to patients' perspectives in planning their diabetes care, providing treatment options to patients, providing strong evidence-based rationale for the need to adopt health behaviors, supporting patients' initiatives regarding their diabetes management, and minimizing the use of controlling language, patients are likely to report significant improvements in their glycemic and psychological outcomes [50]. According to the SDT, autonomy support facilitates intrinsic motivations that drive patients to make decisions that are beneficial to their health outcomes. Interventions that are aimed at increasing patients' perception of autonomy support have also demonstrated effectiveness in improving health behaviors and psychological health outcomes [17,22]. These findings align with the results from the current study which indicate that patients who experience greater perceived autonomy support have better diabetes self-management behaviors, and lower depressive and anxiety symptoms. Additionally, findings from observational studies among adults with T2D are consistent with our results [51–54]. For instance, a recent study among 474 United States adults with T2D found that collaboration between patients and healthcare professionals using autonomy support was associated with improvements in adherence to diabetes self-management behaviors [55]. Studies that use structural equation modeling have shown that perceived autonomy support improves diabetes-related outcomes through improvement in perceived competence and patient satisfaction with care. Autonomy support from healthcare professionals has the potential to motivate patients to "take control" over their diabetes management [55].

In line with our hypothesis, perceived autonomy support ameliorated the negative association between diabetes stigma and self-management. The Johnson-Neyman analysis also revealed that for patients with low perceived autonomy support, the adverse effect of diabetes stigma on self-management was stronger. Greater autonomy support from healthcare professionals can function as a form of social support, empowering patients to effectively cope with diabetes-related stressors [56,57]. This enhanced coping capacity may protect against the adverse effect of stressors (including stigma) on patients' ability to engage in and sustain beneficial self-management behaviors. This finding is consistent with a previous study that found that perceived autonomy support from family and healthcare providers buffered the negative effect of diabetes-related distress on HbA1c levels [57].

The findings from this study have several practical and research implications. First, given the preliminary evidence showing that perceived autonomy support may protect against the negative effects of diabetes stigma, it is important for clinicians in Ghana to ensure that their interactions with adults with T2D are autonomy supportive. This may be challenging given the collectivist and hierarchical nature of Ghanaian society. For instance, studies in Ghana have shown that patient-provider interactions are often characterized by cultural norms that discourage patients from asking questions about their treatment [58] and seeing the physician as their "God [who] decides everything for them" [59,60]. Supporting autonomy of adults with T2D does not mean allowing patients to make medical decisions without any interference or advice from the clinician. Rather, clinicians can ensure autonomy support by actively involving patients, giving advise without the pressure on patients to accept the advice, and ensuring patients have time to carefully consider new information to be able to decide for themselves [61]. There is the need for clinicians to recognize that patients have the ultimate responsibility to make decisions about their diabetes care. Studies have shown that when patients make decisions out of their own volition as opposed to being instructed to act by their clinician in an authoritative way, there is a greater likelihood of better health outcomes [61]. Second, previous research on parental autonomy support in the Ghanaian context have identified two components of autonomy support, i.e., perspective taking (acknowledging patients' viewpoint) and allowance of decision making, indicating that allowing children to make their decision may be less meaningful and impactful on psychological health outcomes within the collectivist Ghanaian culture [23]. Future studies should explore how applicable these components of autonomy support are in the Ghanaian healthcare settings to allow for more nuanced understanding of the impact of perceived autonomy support and to inform future interventions that may enhance perceived autonomy support among adults with T2D. Although the Marbell-Pierre et al. study focused on parental autonomy support, it provides some

preliminary evidence regarding how useful perceived autonomy support may be within the Ghanaian context and may possibly explain the lack of significant moderation effect of perceived autonomy support with respect to psychological outcomes in our study. Third, future studies should also aim to understand the mechanism through which perceived autonomy support moderates the effects of diabetes stigma. Additional studies are also required to understand why perceived autonomy support did not moderate the association between diabetes stigma and psychological outcomes.

The findings of this study should be interpreted in light of some limitations. There is a potential for social desirability bias, especially in assessing perceived autonomy support among participants at the hospital site. Given the collectivistic and hierarchical culture that emphasizes reverence to authority, participants might have felt obliged to rate their healthcare providers favorably to avoid offending them. This could have resulted in the overestimation of perceived autonomy support among study participants. The cross-sectional design also limits our ability to make causal inferences. Future studies should use longitudinal designs to investigate T2D stigma and its effect on behavioral and psychological outcomes over time. The use of non-probability sampling techniques may have introduced selection bias. Additionally, the first author (SA) forward translated all questionnaires to the local Ghanaian dialect, Twi. However, we did not go through the full rigor of translation, back-translation, cultural adaptation, expert review, re-validation, and piloting of instruments due to resource and time constraints. Moreover, while our sample size was adequate, the recruitment of participants from a single tertiary hospital may have limited the generalizability of our findings to all adults with T2D in Ghana. Despite these limitations, this study is one of the first to evaluate perceived autonomy support from healthcare professionals among adults with T2D in Africa highlighting its protective effects against diabetes-related stigma and setting the foundation for SDT-informed interventions to improve diabetes outcomes in Ghana.

## Supporting information

**S1 Checklist. Inclusivity in global research.**
(DOCX)

**S1 Data. Dataset.**
(XLSX)

## Acknowledgments

We thank Dorothy Wilson, Regina Abekah, and Florence Agyapong for their invaluable role in participant recruitment and data collection.

## Author contributions

**Conceptualization:** Samuel Akyirem, Katie Wang, Gail Melkus, Soohyun Nam, Frank Micah, LaRon E Nelson.

**Data curation:** Samuel Akyirem, Emmanuel Ekpor.

**Formal analysis:** Samuel Akyirem.

**Funding acquisition:** Samuel Akyirem.

**Investigation:** Samuel Akyirem, LaRon E Nelson.

**Methodology:** Samuel Akyirem, Emmanuel Ekpor, LaRon E Nelson.

**Project administration:** Samuel Akyirem.

**Supervision:** Katie Wang, Gail Melkus, Soohyun Nam, Frank Micah.

**Validation:** Samuel Akyirem.

**Visualization:** Samuel Akyirem.

**Writing – original draft:** Samuel Akyirem.

**Writing – review & editing:** Samuel Akyirem, Katie Wang, Gail Melkus, Soohyun Nam, Frank Micah, Emmanuel Ekpor, LaRon E Nelson.

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
