## [Decision Letter · Decision Letter 0]

26 Dec 2025

PGPH-D-25-03130

Investigating the role of perceived autonomy support in moderating the association between diabetes stigma and psychological and diabetes self-management outcomes among adults with type 2 diabetes in Ghana.

Dear Dr. Akyirem,

Thank you for submitting your manuscript to PLOS Global Public Health. After careful consideration, we feel that it has merit but does not fully meet PLOS Global Public Health’s publication criteria as it currently stands. Therefore, we invite you to submit a revised version of the manuscript that addresses the points raised during the review process.

The manuscript has been evaluated by three reviewers, and their comments are available below and in the attached document.

The reviewers have commented on aspects such as sampling and participant recruitment, and discussion of the study’s limitations.

Could you please carefully revise the manuscript to address all comments raised?

We look forward to receiving your revised manuscript.

Kind regards,

Ilse Bloom

Staff Editor

Journal Requirements:

2. Please amend your online Financial Disclosure statement. If you did not receive any funding for this study, please simply state: “The authors received no specific funding for this work.”

3. Please update your online Competing Interests statement. If you have no competing interests to declare, please state: “The authors have declared that no competing interests exist.”

4. In the online submission form, you indicated that “The data is part of a larger study from which other manuscripts are currently being developed. We are only able to share data with others upon reasonable request to the corresponding author.”.

a) In a public repository,

b) Within the manuscript itself, or

d) Uploaded as supplementary information.

For further assistance, you may go to: http://journals.plos.org/globalpublichealth/s/data-availability

5. We do not publish any copyright or trademark symbols that usually accompany proprietary names, eg (R), (C), or TM (e.g. next to drug or reagent names). Please remove all instances of trademark/copyright symbols throughout the text, including ® on page 19.

Additional Editor Comments (if provided):

Reviewers' comments:

Reviewer's Responses to Questions

**Comments to the Author**

1. Does this manuscript meet PLOS Global Public Health’s publication criteria?

Reviewer #1: Yes

Reviewer #2: Yes

Reviewer #3: Yes

2. Has the statistical analysis been performed appropriately and rigorously?

Reviewer #1: I don't know

Reviewer #2: Yes

Reviewer #3: Yes

3. Have the authors made all data underlying the findings in their manuscript fully available (please refer to the Data Availability Statement at the start of the manuscript PDF file)?

Reviewer #1: Yes

Reviewer #2: No

Reviewer #3: Yes

4. Is the manuscript presented in an intelligible fashion and written in standard English?

Reviewer #1: Yes

Reviewer #2: Yes

Reviewer #3: Yes

Reviewer #1: I understand that perceived autonomy support will be examined as a moderator in the relationship between diabetes-related stigma and its behavioral and psychological correlates. Overall, the paper is well structured and the information is clearly presented, though several improvements are needed. Please refer to the attached document titled “Review” for the detailed comments.

Reviewer #2: The study investigates perceived autonomy support, a concept from Self-Determination Theory (SDT), as a potential moderator to mitigate the adverse effects of diabetes stigma on psychological and self-management outcomes. The authors' efforts in conducting the research and drafting the manuscript are much appreciated. However, the manuscript still needs improvement in many aspects, as I highlighted below.

Introduction

• Emphasize that this is one of the first studies to explore autonomy support in the context of diabetes stigma in Ghana, a collectivist society (If so). This would underscore the study's unique contribution.

• The manuscript briefly mentions the hierarchical and collectivist nature of Ghanaian society. Expanding on how this cultural context might influence autonomy support and diabetes stigma could provide a richer understanding of the research.

• “Whereas autonomy support is universal, it may not present in the same manner and may not be as beneficial across cultures (19)”. While the authors mention across cultures, there should have been many references from different cultures.

Method

• While the manuscript mentions purposive and convenience sampling, it could provide more details about how participants were selected and whether this approach might introduce bias. For example, were there any efforts to ensure diversity in the sample (e.g., age, gender, socioeconomic status)? This has to be stated in the limitation section. Meanwhile, I suggest the authors interpret the socioeconomic characteristics of the study population in light of those of studies from other parts of the world with diverse cultures (particularly from high-income countries). This should have been added in the discussion section by using the following suggested articles.

https://doi.org/10.1155/2020/9408316

https://doi.org/10.1016/j.clinthera.2016.07.006

https://doi.org/10.2337/ds23-0013

• The manuscript states that surveys were administered via the Qualtrics mobile app. It would be helpful to clarify whether the app was used for both researcher-administered and self-administered surveys, and how data quality was ensured during these processes.

• The statistical methods are well-described, but the rationale for choosing hierarchical multivariable linear regression and the Johnson-Neyman technique could be briefly explained for readers unfamiliar with these methods.

• While ethical approval and informed consent are mentioned, the manuscript could elaborate on how participant confidentiality and data security were maintained during the study.

Results

“Table 1 shows descriptive statistics.” This should be Table 2.

Table 2: “Perceived autonomy support was negatively and significantly associated with anxiety (r=-0.23) and depressive symptoms (r=-0.27). Participants who reported greater perceived autonomy support tend to engage in better diabetes self-management (r=0.24).” The correlation r<0.3 is usually considered weak. The authors should state this in the text as well.

Table 3: p-value of DSAS-2 in block 2 written as <0.00001, and this should be corrected as <0.0001. Please check that all the decimals are consistent.

Figures: Ensure figures are high-resolution for better readability, especially for publication. Use consistent font sizes, styles, and colors across all figures for a professional appearance. The captions should provide a brief explanation of the figure's content, including the key findings and any critical annotations.

Figure 1

Clarity of Labels: The axis labels are not fully descriptive. For example, "Diabetes-related stigma" could be labeled as "Diabetes Stigma (DSAS-2 Score)" for clarity. Similarly, "Anxiety symptoms" and "Depressive symptoms" could be labeled with their respective scales (e.g., "Anxiety Symptoms (PROMIS Anxiety T-Score)" and "Depressive Symptoms (PROMIS Depression T-Score)").

Data Points: The scatter plot appears cluttered, making it difficult to interpret the relationships. Consider reducing the size of the data points or using transparency to make overlapping points more distinguishable.

Trend Lines: The trend lines are present, but their labels or legends are missing. Adding a legend or annotations to clarify the meaning of the trend lines would improve readability.

Figure 2

Axis Labels: The x-axis label "T2D stigma z-scores" could be expanded to "Diabetes Stigma (Standardized Z-Scores)" for clarity. The y-axis label "Autonomy support" is unclear. It could be updated to "Interaction Effect with Diabetes Stigma" to reflect better the data being presented.

Legend: The figure lacks a legend explaining the meaning of the different lines (e.g., "Mean z-score," "1 SD," etc.). Adding a legend would make the figure more straightforward to interpret.

Visual Design: The figure could benefit from a more apparent distinction between the shaded areas (e.g., using different colors or patterns for "n.s." and "p < .05"). The dashed vertical line could be labeled to indicate its significance (e.g., "Threshold for significant moderation effect").

Figure 3

Axis Labels: The x-axis label "Autonomy Support (z-score)" is clear. Still, the y-axis label "Interaction Effect with Diabetes Stigma" could be expanded for clarity (e.g., "Interaction Effect of Autonomy Support on Diabetes Stigma and Self-Management").

Legend: The legend is present but could be more descriptive. For example, "n.s." could be expanded to "Not Significant," and "p < .05" could be labeled as "Significant Interaction Effect."

Range of Observed Data: The range of observed data is marked, but it would be helpful to include a brief explanation of its significance in the figure caption.

Visual Design: The shaded area for "p < .05" could be made more distinct to improve visual clarity.

Discussion

The Discussion section of the manuscript is well-written and provides a comprehensive analysis of the findings. However, there are areas where improvements can enhance clarity, address limitations more thoroughly, and strengthen the connection to the broader research context.

Strengthen the Connection to Broader Research

• Highlight the novelty: Emphasize the study's contribution to the limited research on diabetes stigma in low-middle income countries and the role of autonomy support in mitigating its effects.

• Compare with global findings: Discuss how the findings align or differ from studies conducted in high-income countries, and what this means for global diabetes care strategies.

Address Limitations More Thoroughly

• Expand on social desirability bias: Discuss how self-reported measures might have influenced the results, particularly in a collectivist society like Ghana, where individuals may feel pressured to provide socially acceptable responses.

• Cross-sectional design: Highlight the inability to establish causality and suggest future longitudinal studies to confirm the causal relationships between autonomy support, stigma, and diabetes outcomes.

• Sample size and generalizability: Discuss whether the sample size (190 participants) and the recruitment from a single tertiary hospital in Ghana may limit the generalizability of the findings to other populations or settings.

Reviewer #3: 1) Title and Abstract

The topic is appropriate, and the abstract is clearly written. It includes the essential elements of a scientific abstract, namely the study design, setting, sample size, key variables, analytic approach, and principal findings. However, the abstract implies moderation of both behavioral and psychological outcomes, whereas moderation was demonstrated only for diabetes self-management and not for psychological outcomes. This discrepancy should be addressed to ensure consistency between the abstract and the reported results.

2) Introduction and Background

Strengths

• A strong epidemiological rationale is provided using IDF and GBD estimates.

• Diabetes-related stigma in Ghana is clearly described, including culturally relevant manifestations.

• The application of Self-Determination Theory (SDT) is appropriate and well articulated.

• The study hypotheses are clearly stated and aligned with the proposed moderation analyses.

Comments

The authors state that autonomy support has not been investigated in the diabetes-related stigma literature. This claim should be made more cautiously unless supported by a formal scoping or systematic review

3) Methods

Strengths

• The study employs well-established and validated instruments (HCCQ, DSAS-2, PROMIS, PAID, DSMQ).

• The overall statistical approach, including moderation analysis, is appropriate for the research questions.

Comments

• The use of purposive and convenience sampling should be justified. While non-probability sampling is common in exploratory and observational research, its rationale should be explained, and implications such as selection bias and limited generalizability should be explicitly discussed.

• Details of participant recruitment are insufficient. The manuscript should report how many patients were approached, how many were eligible, how many declined participation, and how many were ultimately included. If available, any systematic differences between participants and non-participants should be noted.

• The role and training of data collectors are not adequately described.

• Although questionnaires were administered verbally in Twi, details regarding translation procedures (e.g., forward–backward translation, piloting, or cultural adaptation) are missing. The absence of this information represents a significant threat to measurement validity and should be addressed or acknowledged as a limitation.

• Participant compensation, while ethically approved, should be briefly discussed as a potential source of participation bias.

• For PROMIS depression and anxiety measures, the manuscript describes conversion from raw scores to T-scores, but it is unclear whether raw scores or T-scores were used in analyses and tables. This should be clearly specified.

4) Results and Tables- Comments

• There is an inconsistency in the reported duration of diabetes (range 1–55 years in Table 1 versus 1–50 years elsewhere), which should be corrected.

• Table numbering and referencing are inconsistent: the text indicates that Table 1 contains bivariate associations, whereas these are presented in Table 2.

• The classification of “problematic stigma” as one standard deviation above the mean requires justification through a clear operational definition and rationale, particularly given the absence of a widely accepted clinical cut-off.

5) Discussion and Conclusion

Strengths

• The discussion appropriately acknowledges that moderation was observed only for diabetes self-management.

• The findings are interpreted within the sociocultural context of Ghana in a thoughtful and nuanced manner.

**Do you want your identity to be public for this peer review?** For information about this choice, including consent withdrawal, please see our Privacy Policy

Reviewer #1: No

Reviewer #2: **Yes:** Dr. Palanisamy Amirthalingam

Reviewer #3: **Yes:** Kalaiselvan Ganapathy

---

## [Decision Letter · Decision Letter 1]

25 Jan 2026

PGPH-D-25-03130R1

Investigating the role of perceived autonomy support in moderating the association between diabetes stigma and psychological and diabetes self-management outcomes among adults with type 2 diabetes in Ghana.

Dear Dr. Akyirem,

Thank you for submitting your manuscript to PLOS Global Public Health. After careful consideration, we feel that it has merit but does not fully meet PLOS Global Public Health’s publication criteria as it currently stands. Therefore, we invite you to submit a revised version of the manuscript that addresses the points raised during the review process.

To aid the reviewers in evaluating the revisions, could you please edit your "response to reviewers" file to indicate using line numbers where each revision has been added. Please note there is no need to re-upload the other revised files (e.g. figures).

We look forward to receiving your revised manuscript.

Kind regards,

Alejandro Torrado Pacheco, PhD

Staff Editor

Journal Requirements:

Additional Editor Comments (if provided):

Reviewers' comments:

Reviewer's Responses to Questions

**Comments to the Author**

Reviewer #1: All comments have been addressed

Reviewer #2: All comments have been addressed

Reviewer #3: All comments have been addressed

publication criteria?

Reviewer #1: Yes

Reviewer #2: Yes

Reviewer #3: Yes

3. Has the statistical analysis been performed appropriately and rigorously?

Reviewer #1: Yes

Reviewer #2: Yes

Reviewer #3: Yes

4. Have the authors made all data underlying the findings in their manuscript fully available (please refer to the Data Availability Statement at the start of the manuscript PDF file)?

Reviewer #1: Yes

Reviewer #2: Yes

Reviewer #3: Yes

5. Is the manuscript presented in an intelligible fashion and written in standard English?

Reviewer #1: Yes

Reviewer #2: Yes

Reviewer #3: (No Response)

Reviewer #1: The authors have responded appropriately to the comments on their manuscript and and have corrected formatting errors.

Reviewer #2: I appreciate the authors for their revision.

Unfortunately, I didn't find the figures (1-3) in the revised file. I couldn't check the revised figures as I suggested more comments in my forst review. Therfore, I ask the authors to submit the manuscript once again with the revised files.

In revised manuscript, it is very difficult to find out the revision done by the authors. I recommend the authors to provide the response with page number and line number. So that it provides clear idea to reviewer that where the revision took place in the manuscript exactly.

Reviewer #3: The author has responed to the Comment 3:But a statment on the puropos of selection the sampling strategy can be included.

**Do you want your identity to be public for this peer review?** For information about this choice, including consent withdrawal, please see our Privacy Policy

Reviewer #1: **Yes:** Lucie Venet-Kelma

Reviewer #2: **Yes:** Palanisamy Amirthalingam

Reviewer #3: **Yes:** Dr.Kalaiselvan Ganapathy, Professor, Department of Community and Family Medicine, All India Institute of Medical Sciences, Mangalagiri, Andhra Pradesh, India

---

## [Decision Letter · Decision Letter 2]

4 Feb 2026

Investigating the role of perceived autonomy support in moderating the association between diabetes stigma and psychological and diabetes self-management outcomes among adults with type 2 diabetes in Ghana.

PGPH-D-25-03130R2

Dear Mr Akyirem,

We are pleased to inform you that your manuscript 'Investigating the role of perceived autonomy support in moderating the association between diabetes stigma and psychological and diabetes self-management outcomes among adults with type 2 diabetes in Ghana.' has been provisionally accepted for publication in PLOS Global Public Health.

Best regards,

Julia Robinson

Executive Editor

Reviewer Comments (if any, and for reference):

Reviewer's Responses to Questions

**Comments to the Author**

Reviewer #2: (No Response)

publication criteria?

Reviewer #2: Yes

3. Has the statistical analysis been performed appropriately and rigorously?

Reviewer #2: Yes

4. Have the authors made all data underlying the findings in their manuscript fully available (please refer to the Data Availability Statement at the start of the manuscript PDF file)?

Reviewer #2: Yes

5. Is the manuscript presented in an intelligible fashion and written in standard English?

Reviewer #2: Yes

Reviewer #2: I appreciate the authors for addressing all the comments. I have no further queries.

**Do you want your identity to be public for this peer review?** For information about this choice, including consent withdrawal, please see our Privacy Policy

Reviewer #2: **Yes:** Palanisamy Amirthalingam
